# NoiseOut: Learning to Gate Improves Robustness in Deep Neural Networks

## Abstract

Deep Neural Networks (DNNs) achieve impressive performance, when trained on datasets of similar distributions. However, they struggle to generalize to novel data, such as image perturbations, when they differ from the training distribution. Using the Integrated Gradients method, we visualize several perturbed features contributing to the higher classification errors. To filter out such distractor features, we take inspiration from the thalamus, which is a biological gating mechanism that improves the signal fidelity of novel stimuli for task completion. Similarly, we propose a novel method called NoiseOut which is a lightweight modular gating mechanism that can be easily integrated with existing DNNs to enhance its robustness to novel image perturbations. When training on the clean datasets, we randomly replaced a subset of the hidden states with normally-sampled values and, augmented the Integrated Gradients analysis method into an additional objective function. With these processes, NoiseOut gradually learned suitable dynamic gating policies to filter out distractor signals and pass task relevant information to the classifier. When evaluating on perturbed datasets, NoiseOut uses the prior learned gating policies to filter out features that negatively influence classification. We demonstrate that our modular NoiseOut mechanism improves existing DNN's robustness to novel perturbations by achieving strong results on the MNIST-C and ImageNet-C benchmarks. [1]

## 1 Introduction

Machine learning algorithms, especially Deep Neural Networks (DNNs), struggle to generalize beyond their training data if the novel data points are sufficiently dissimilar to the training distribution. Unfortunately, this is a common occurrence in real-world applications. For example, the lighting conditions or the camera used when taking a photo might differ from those for the training dataset.

A number of benchmarks, such as MNIST-C Mu & Gilmer (2019) and ImageNet-C Hendrycks & Dietterich (2018), have been established to measure the ability of models to generalize to novel perturbations. On the ImageNet-C benchmark Hendrycks & Dietterich (2018), the authors showed that the test set performance of a ResNet-50 model degrades from $76.15\%$ to $39.17\%$ on the perturbed test set.

Existing solutions can be sub-divided into three main categories, namely, 1) input data augmentation Guertler et al. (2022); Müller et al. (2019); Sundaram & Hulkund (2021); You et al. (2018); Zhong et al. (2017), 2) model adjustments during training Srivastava et al. (2014); Wan et al. (2013) and 3) pre-training the models on vast amounts of data with different distributions Chen et al. (2023); Devlin et al. (2018); Dosovitskiy et al. (2020); Hu et al. (2019). All three methods run orthogonal to one another and can thus be used at the same time.

The method proposed in this work, coined *NoiseOut*, is designed to jointly learn a gating mechanism that a model can use to filter out irrelevant information during training and inference. To teach the NoiseOut mechanism which information to ignore, during the training phase, we randomly replace a subset of the hidden states with normally-sampled values, where the average and standard deviation are based on the distribution of the feature to be replaces within the current batch. In theory, this

---

[1] All code will be made available upon paper acceptance.

teaches the network to ignore distractor signals and focus only on reliable information to make a prediction.

Our contributions include:

1. We introduce a selective gating module that learns to gate task relevant information in a dynamic manner; and can easily be integrated with existing deep learning models.

2. We introduce Noise Injection that rather than adding noise to hidden states, replaces nodes with randomly sampled values that mimick the real values, making it harder for the Gating mechanism to decide what information to gate.

3. We show the efficacy of the gating mechanism in the latent space by visualizing the Integrated Gradients Sundararajan et al. (2017), and determine what aspects of the input image influence the gating.

4. We demonstrate that the proposed method achieves strong performances on MNIST-C, and on ImageNet-C when used by itself and in conjunction with existing methods, respectively.

The remainder of the paper is structured as follows: Section 2 highlights techniques to improve out-of-distribution predictions. Section 3 provides a detailed description of the model architecture. In Section 4, we test the architecture in conjunction with the current state-of-the-art approaches on various robustness benchmarks, and conclude in Section 5.

## 2  RELATED WORK

### 2.1  NOISE INJECTION AND GATING MECHANISMS

Dropout Srivastava et al. (2014) and DropConnect Wan et al. (2013) are among the most popular techniques to improve deep learning's robustness and generalization capabilities, wherein a subset of neurons or weights are randomly zeroed during training. The idea is to prevent overfitting and encourage the learning of more robust features. Instead of setting neuron activation to zero or adding noise to the hidden states You et al. (2018), our proposed NoiseOut method replaces state values with randomly sampled values.

The proposed solution is inspired by the selective gating mechanism played by the basal ganglia. It has been argued that the thalamus modulates information flow to the prefrontal cortex Alexander et al. (1986); Wickens (1997); Wimmer et al. (2015) by disinhibiting task relevant information and inhibiting distractor stimuli. The prefrontal cortex maintains Kumar et al. (2022); Parthasarathy et al. (2019) the modulated information for task completion. It has been postulated that a suitable gating policy can be learned by first randomly sampling inhibition or disinhibition actions and reinforcing the correct actions based on an objective function Frank (2005); Frank et al. (2001); O'Reilly & Frank (2006). Simple computational models can gradually learning a suitable gating mechanism to solve distractor based tasks Lloyd et al. (2012); more importantly, the gradually learned gating policy can rapidly generalize to novel stimulus combinations Kumar et al. (2023), suggesting gating could be a meta-learning mechanism.

LSTMs Hochreiter & Schmidhuber (1997) internally employ a gating mechanism to control the flow of information over time, improving performance on tasks with long-term dependencies. Attention mechanisms Vaswani et al. (2017) also utilize a form of gating, where the importance of each input is learned and used to weight the contribution of that input to the output. However, adding a standalone gating mechanism to existing deep learning architectures has not been explored.

### 2.2  DATA AUGMENTATION AND PRE-TRAINING MODELS

Data augmentation techniques generate additional training examples by applying various transformations to the original data Guertler et al. (2022); Müller et al. (2019); Sundaram & Hulkund (2021); Zhong et al. (2017). The model, therefore, gets exposure to a wider distribution of data variations, improving its robustness and ability to generalize to novel instances. While data augmentation works on the input side, our proposed NoiseOut mechanism operates on the hidden representations, providing a complementary representation augmentation approach.

Pre-training models on various large datasets allows models to learn generic features from different data distributions. Subsequent fine-tuning on a specific dataset allows these models to use the general features to learn specific representations for improved generalization performances Chen et al. (2023); Devlin et al. (2018); Dosovitskiy et al. (2020); Hu et al. (2019). Our NoiseOut method, in contrast, does not rely on additional data but instead focuses on leveraging the model's internal representation for improved robustness.

## 3 MODEL

### 3.1 THE NEED TO GATE INFORMATION

Before introducing our method in detail, we offer some further motivation for the architecture. To do so, we trained a simple CNN classifier [2] on the clean MNIST trained set, and tested it on the perturbed images of the MNIST-C test set (more information about this dataset can be found in Section 4).

To determine which hidden states of the CNN contribute positively to the correct prediction, and which are distracting the model, we use the Integrated Gradient method Sundararajan et al. (2017) to interpret which input variables and features contribute to the model predictions. We accomplish this by integrating gradients of the model's output with respect to its input features along a straight path, providing a visual representation of how each feature contributes to the prediction.

$$\text{IG}_i\left(x\right) = \left(x_i - x_i'\right) * \int_{\alpha=0}^{1} \frac{\partial F\left(x' + \alpha * \left(x - x'\right)\right)}{\partial x_i} d\alpha \tag{1}$$

where $x'$ (initialized as a zero vector in our experiments) is the baseline for the straight path ending at $x$, along which the gradients are calculated. For more details we refer the reader to Sundararajan et al. (2017). This can be estimated by

$$\text{IG}_i\left(x\right) \sim \left(x_i - x_i'\right) * \frac{1}{m} \sum_{k=1}^{m} \frac{\partial F\left(x' + \frac{k}{m} * \left(x - x'\right)\right)}{\partial x_i} \tag{2}$$

where we use $m = 50$ in all of our experiments (which we found to empirically work best and is within the recommended range of $[20, 300]$ that is mentioned in Sundararajan et al. (2017)).

Since our datasets have a finite number of classes, if some feature contributes negatively to the probability of the seen image being of a class that isn't the target class, we can indirectly argue that the positively contributing feature signals the correct prediction. Thus, we augment the calculation to

$$\text{FIG}_i\left(x\right) = \sum_{j=0}^{C} \left(1 - 2 * \mathbb{1}_{\{j \neq \text{target}\}}\right) * \left(x_i - x_i'\right) * \frac{1}{m} \sum_{k=1}^{m} \frac{\partial F\left(x' + \frac{k}{m} * \left(x - x'\right)\right)}{\partial x_i} \tag{3}$$

where "FIG" abbreviates "Full Integrated Gradients", $C$ denotes the number of classes and $\mathbb{1}$ denotes the indicator function.

This means that, generally speaking, a positive FIG value indicates that a specific feature (or hidden state) contributes more towards the correct prediction than to the wrong ones, whilst a negative values means the opposite.

Figure 1 provides a side-by-side comparison of the Estimated Full Integrated Gradients for both the input image, and the first hidden layer after flattening (the latter has been re-shaped for easier visualization). Even though the network was trained until convergence, the model is unable to filter out information in both the clean and Brightness perturbed images, contributing to wrong predictions (indicated by the dark spots). This leads to a test performance drop from 99.17% accuracy on the clean test set to 52.42% accuracy on the Noise perturbed test set. Note that the color scales for the two FIG (hidden) plots are not the same.

Figure 2 shows the input and hidden state FIG distributions where there is a clear increase in negative values on the Brightness perturbed test image compared to the clean test image. Therefore,

---

[2]Unless specified otherwise, we use the architecture, hyperparameters and training procedure of https://github.com/pytorch/examples/tree/main/mnist

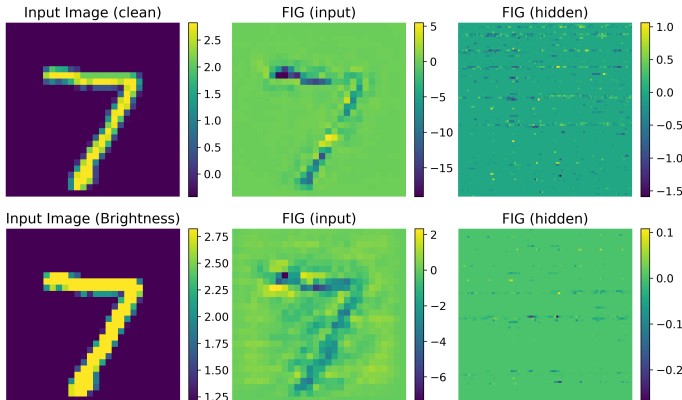

Figure 1: Full Integrated Gradients from a pretrained CNN when given a clean (top) and a brightness perturbed (bottom) test image. Features from the input (center) and the first hidden layer after flatting (right) contributing positively (above 0) and negatively (below 0) to predict digit "7" can be visualized.

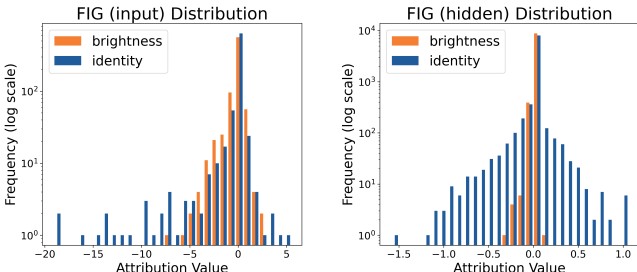

Figure 2: Brightness perturbed test images increases negative FIG values compared to the clean test images. Note the exponential Y axis.

a mechanism that gates information that contributes negatively to the prediction could improve robustness to perturbations.

## 3.2 THE NOISEOUT ARCHITECTURE

Figure 3 shows the NoiseOut architecture with two important components, a) an explicit gating mechanism that depends on the hidden layer activity, and b) a noise injected training regime that teaches the network which information to gate. Figure 9 in Appendix C shows how the NoiseOut mechanism can be easily integrated into a standard CNN architecture.

### 3.2.1 IMPLEMENTATION OF THE GATING MECHANISM

The gates for a specific hidden layer are implemented based on the values of a single hidden layer.

$$\Omega = \sigma \left( h_i * W + b \right) \tag{4}$$
$$h_i' = h_i \odot \Omega \tag{5}$$

Where $\odot$ denotes the Hardmard (element-wise) product, $h_i$ are the elements of the i-th hidden layer, $\sigma$ is the Sigmoid activation function and $W, b$ are trainable parameters.

### 3.2.2 LEARNING TO GATE USING NOISE INJECTION

To learn a suitable gating policy during training, we intentionally introduce distractor information ($z_i$) into the hidden layer ($h_i$). To do this, we randomly select a subset of the hidden states using a

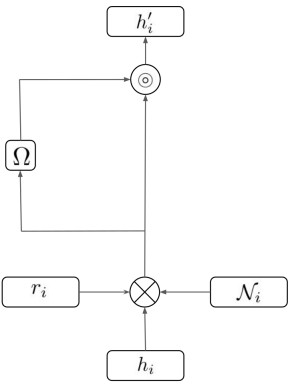

Figure 3: NoiseOut gating architecture

Bernoulli distribution $r_i^{j,k} \sim$ Bernoulli$(p)$ with $p$ as the probability of selecting a given hidden state, and then replace the hidden state values with normally-generated random values according to:

$$\mathcal{N}_i = \text{Normal}\left(\frac{1}{B}\sum_{n=1}^{B}h_i^{n,k}, \sqrt{\frac{1}{B}\sum_{n=1}^{B}\left(h_i^{n,k} - \bar{h}_i^k\right)^2}\right) \tag{6}$$

$$z_i^{j,k} \leftarrow h_i^{j,k} \cdot (1 - r_i^{j,k}) + r_i^{j,k} \cdot \mathcal{N}_i \tag{7}$$

where $z_i^{j,k}$ is the $k$-th hidden state of the $i$-th layer and $j$-th sample in the batch, $h_i^{j,k}$ is the original value of the hidden state, $B$ is the batch size, $\bar{h}_i^k$ is the mean value of the hidden states along the batch dimension, and Normal$(\mu, \sigma)$ is a normal distribution with mean $\mu$ and standard deviation $\sigma$.

The mean and standard deviation of the normally-generated values in Eq. 7 are estimated along the batch dimension, to ensure that the exchanged values have similar statistical properties as the original hidden states. All other variables and loss functions are kept the same as the usual training regime.

### 3.3 ADDITIONAL OBJECTIVE FUNCTION BY INTEGRATING THE INTEGRATED GRADIENTS

Besides the proposed gating mechanism, we develop an additional loss term that can be added to the standard categorical loss term used to train a classifier (right side of Eq. 3).

The additional objective function to be minimized is constructed using a binarized version of the Full Integrated Gradients, and this provides two advantages. Firstly, it encourages the gates to return binary outputs, which effectively reduces the bleeding of information, and secondly, it provides an explicit target for the gating, rather than computing the gradients for the gate implicitly using the categorical loss function.

The binarization can be achieved using a step function which sets all values lower than 0 to 0 and all values greater than 0 to 1.

$$\text{BinarizedFIG}\left(\text{FIG}\left(x_i\right)\right) = \begin{cases} 1 & \text{if } x_i > 0 \\ 0 & \text{if } x_i \leq 0 \end{cases} \tag{8}$$

with FIG $(x_i)$ is estimated via Equation 3.

$$\mathcal{L} = \frac{1}{N}\sum_{i=0}^{N}\sum_{j=0}^{M} -y_j \log\left(\hat{y}_{\theta,i}\right) + \lambda * \frac{1}{N}\sum_{i=0}^{N}(\mathcal{A} - \mathcal{B})^2 \tag{9}$$

where

- $M$ and $N$ are the number of classes and number of elements in the batch respectively

- $\mathcal{A} = \text{BinarizedFIG}\left(\text{FIG}\left(\text{hidden}_{\theta,i}\right)\right)$
- $\mathcal{B} = \sigma\left(\text{hidden}_{\theta,i} * W + b\right)$
- $\lambda$ is a hyperparamter that determines the influence the FIG loss has.

We empirically determined that $\lambda = 0.5$ works best (See Table 3 in Appendix B), and will use it in all experiments, unless specified otherwise. To the best of our knowledge, this is the first time the Integrated Gradients is used as an add-on objective function besides as an analysis tool.

## 4 EXPERIMENTS

To demonstrate the robustness afforded by NoiseOut, we test it on the MNIST-C and ImageNet-C robustness benchmarks, either as a standalone model or as an add-on to other deep learning models. For all experiments, models were initially trained on the clean training set and subsequently evaluated on different version of the test set, each one augmented with a different perturbation.

### 4.1 TRAIN ON MNIST AND EVALUATE ON MNIST-C

All models assessed on the MNIST-C Mu & Gilmer (2019) benchmark were trained on the clean MNIST dataset with 60,000 images and tested on 16 different version of the perturbed test set (See Figure 6 in Appendix A.1 for perturbation examples). The authors benchmarked a plethora of state-of-the-art robustness methods Frosst et al. (2018); Madry et al. (2017); Schott et al. (2018); Wang & Yu (2019) against a standard CNN [3]. Interestingly, the CNN outperformed the more sophisticated models. Thus, for this benchmark, we will compare NoiseOut to a CNN with the same architecture. The NoiseOut and NoiseOut FIG models adopt the standard CNN model with modifications following section 3.2, and section 3.2 coupled with section 3.3 respectively.

| Noise Injection | 0% | 10% | 20% | 30% |
|---|---|---|---|---|
| CNN | $87.43\% \pm 1.29$ | $87.39\% \pm 0.85$ | $87.51\% \pm 1.45$ | $86.07\% \pm 1.53$ |
| NoiseOut | $91.09\% \pm 0.55$ | $91.71\% \pm 0.35$ | $92.03\% \pm 0.37$ | $89.89\% \pm 0.46$ |
| NoiseOut FIG | $91.51\% \pm 0.47$ | $92.05\% \pm 0.39$ | $\mathbf{92.30\%} \pm 0.40$ | $92.19\% \pm 0.43$ |

Table 1: Performance (mean and stdev over 10 runs) achieved on the MNIST-C test set by the standard CNN, NoiseOut and NoiseOut FIG models with different Noise Injection probabilities.

The results demonstrate that our NoiseOut method improves the standard CNN's robustness to novel perturbations in the testing data. As expected, augmenting the loss function with the empirically calculated FIG further improves performance. However, this comes at the cost of additional computations due to the need to perform backpropagation multiple times to compute the gradients. One solution is to train a second neural network to predict the target FIG, similar to estimating the Wasserstein function Arjovsky et al. (2017). Reducing FIG computation during training can be explored in future work.

Replacing hidden state values in the standard CNN model with different probabilities of randomly sampled values did not lead to a significant change in MNIST-C baseline performance, demonstrating that Noise Injection without learning the NoiseOut gating mechanism does not improve model robustness. Conversely, learning to gate with no Noise Injection (0%) and without the additional FIG loss term still led to an increase in model robustness, demonstrating the relevance of having a gating mechanism. Yet, the highest performance was achieved when the standard CNN model used: 1) the NoiseOut gating mechanism to learn relevant gating activity, 2) Noise injection with 20% probability of exchange, and 3) the FIG method to compute an additional objective function.

### 4.1.1 ANALYSIS

We visualized which features were gated away by comparing the Full Integrated Gradients of the hidden layer before and after gating. The network used is the standard CNN model augmented with the NoiesOut mechanism, but trained without the additional FIG loss term from Equation 9 that

---

[3]https://github.com/pytorch/examples/tree/main/mnist

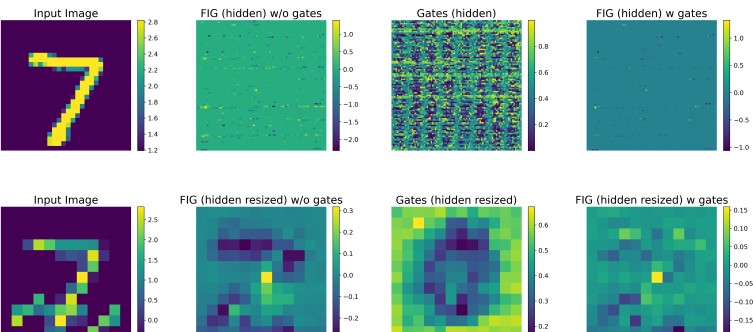

Figure 4: The figure shows the input image and the Full Integrated Gradients (FIG) of the hidden states before and after applying the gating. All plots correspond to a test image of the digit "7", where the top and bottom rows are brightness and zigzag perturbations respectively.

achieved 92.30% in Table 1. Figure 4 (top) shows a clear reduction in negative FIG values in the hidden states after the information is gated by the NoiseOut mechanism.

Since the gating is at the level of the hidden layer, the visualization of the gates do not offer intuitive insights. Hence, the input image was first reshaped via a bi-linear interpolation from it's original shape $(28, 28)$ to $(12, 12)$. Then, the hidden layer and gating activity were reshaped from $(9216,)$ to $(64, 12, 12)$ and the average across the first dimension was taken to generate the final plots of shape $(12, 12)$ to be consistent with the reshaped input image. Though this visualization method might not work in other cases, it is easier to see that most of the distractor information with negative FIG values has been filtered out (Fig. 4 bottom right).

## 4.2 TRAIN ON IMAGENET AND EVALUATE ON IMAGENET-C

Similar to the set-up of MNIST-C, models were first trained on the clean ImageNet training set and evaluated on 15 versions of the ImageNet-C test set, each with a different perturbations (examples of the perturbations can be found in Figure 7 in Appendix A.2).

In this section we will demonstrate that the NoiseOut mechanism can be integrated with existing state-of-the-art (SOTA) models to improve model robustness. To do so, we modify two SOTA models by adding the NoiseOut mechanism to the first hidden layer after flattening the CNN output. The first model, ANT Rusak et al. (2020), is the best ranked standalone model and the second is the current SOTA method which is a combination of DeepAugment Hendrycks et al. (2021) and AugMix Hendrycks et al. (2020).

**ANT Rusak et al. (2020).** A noise generator samples values from a normal distribution and is trained to maximize the classification error while the classifier is trained to minimize the error under different adversarial noise distributions. To maintain high classification accuracy on clean samples, each mini-batch contained 50% clean data and the rest was perturbed with the noise generator's output. The noise generator's architecture was also modified to allow for local spatial correlations, enabling the learning of diverse distributions to increase the model's robustness against image corruptions like rain or snow.

**DeepAugment Hendrycks et al. (2021) + AugMix Hendrycks et al. (2020).** DeepAugment is a data augmentation technique that perturbs internal representations of deep networks while generating semantically consistent images with unique and diverse distortions. A clean image is passed through the image-to-image network and several perturbations are randomly sampled from a set of manually designed functions to modulate the network weights and processed information. Examples of perturbations include zeroing, negating, convolving, transposing, and applying activation functions. This results in semantically consistent images with diverse distortions, which have been shown to outperform other methods on benchmarks such as ImageNet-C and ImageNet-R. AugMix is another data augmentation technique that stochastically samples and layers augmentation operations to generate a high diversity of augmented images. The classifier is trained to learn a consistent embedding of the same input image across diverse augmentations through the use of Jensen-Shannon

divergence as a consistency loss. Mixing DeepAugment and AugMix methods generates diverse transformations of the input images, which is essential to learn robust features and prevent models from memorizing fixed augmentations.

Since both of the above SOTA methods are focused on augmenting the input images during training, incorporating NoiseOut is straightforward as the gating mechanism and Noise Injection can be easily appended to the network at the first hidden layer after flattening the convolution output. Since Noise Injection probability of 20% worked best for MNIST-C, we used the same hyperparameter for the ImageNet-C experiments. Models integrated with NoiseOut were trained with the same hyperparameters as ANT Rusak et al. (2020) and DeepAugment Hendrycks et al. (2021). Furthermore, NoiseOut only attributed to a fractional percentage of the computational load.

The models were assessed based on the Mean Corruption Error (mCE) which is the average of the Corruption Errors (CE) on the perturbed test sets as compared against a vanilla AlexNet model Hendrycks & Dietterich (2018)

$$\text{CE}_c^f = \left( \sum_{s=1}^{5} E_{s,c}^f \right) \bigg/ \left( \sum_{s=1}^{5} E_{s,c}^{\text{AlexNet}} \right) \tag{10}$$

where $f$ is the model used, $s$ denotes the severity of the corruption and $c$ the corruption.

| Model | Mean Corruption Error |
|---|:---:|
| ResNet-50 Baseline | 76.7% |
| ANT | 62.51% |
| ANT + NoiseOut (0.2) | **61.62%** |
| DeepAugment + AugMix | 53.6% |
| DeepAugment + AugMix + NoiseOut (0.2) | **53.4%** |

Table 2: ImageNet-C mean Corruption Error (mCE) achieved by the two SOTA baseline models and the NoiseOut augmented versions. (Lower number means better performance).

As can be seen in Table 2, when SOTA models are combined with NoiseOut, new state-of-the-art robustness performances of 61.62% and 53.4% can be achieved on the ImageNet-C benchmark.

### 4.2.1 ANALYSIS

The gating mechanism learned in Kumar et al. (2023) was consistent despite different stimuli, hence the same gating policy could generalize to novel stimuli. Figure 4 bottom showed that the gates learned to filter out information in the center of the input image but retained information closer to the extremities. The same gating policy might be sufficient and generalize to different digits in MNIST-C since most digits are located in the center of the image. However, learning a single gating policy might not be sufficient especially if objects of interests are in different regions of the image.

Hence, for ImageNet-C, we were curious to know if the prior learned gating policy was the same (i.e. consistently gate specific pixel values) despite the input image; or if the gating activity was specific to each image type, perturbation and severity of perturbation (i.e. different pixels are filtered out depending on the image).

We collated the gating activity across five different images types, five different perturbation effects and five levels of perturbation severity. The gating activity for each condition (image, perturbation, severity) was classified according to the five categorical subclasses using Linear Discriminant Analysis (LDA). The gating activity was projected onto the top two linear components (LDA1 and LDA2) to visualize if the different conditions clustered according to each subclass.

The left plot in Figure 5 shows the clusters for the five different images occupy distinct subspaces, despite the differences in perturbation or severity. This is a clear indication that the gating activity is different for each image. This is expected as the object of interest is likely to be in different regions of the image, hence requiring distinct gating activity to filter out distractor information.

The center plot shows the clusters for Defocus blur and Motion blur as well as Brightness and Pixelate conditions are slightly overlapped, suggesting that related perturbations might cause similar gating

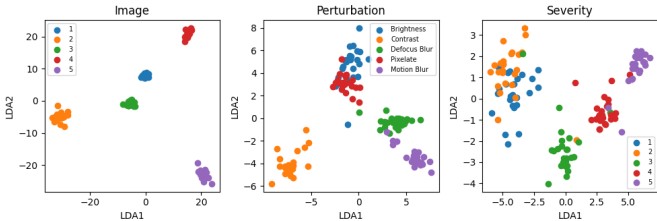

Figure 5: LDA suggests gating activity for ImageNet-C is influenced by different images (left), perturbations (center) and severity (right).

activity. In the right plot, the clusters grouped according to severity are organized from left to right along the LDA1 axis, indicating that the gating activity changes in line with increasing severity.

The tightness of the clusters decrease from the left to right plots, suggesting that the gating activity is highly dependent on the image type as compared to the type of perturbation, and is less dependent on the perturbation severity.

Overall, Figure 5 offers strong evidence that the gating activity is dynamic, as it is starkly influenced by the three different conditions and five categorical subclasses, unlike the single static gating policy learned in Kumar et al. (2023). This underscores NoiseOut's versatility in learning dynamic gating policies instead of static biased gates, to improve the model's robustness in distractor based classification tasks.

## 5 CONCLUSION

In this work, we proposed NoiseOut, a selective gating mechanism for improving the robustness of deep neural networks to novel perturbations in input data. During the training phase, instead of passing novel perturbed information directly to a classifier, NoiseOut gradually learns to filter out distractor information and passes the task relevant information for classification. During evaluation, the model focuses on reliable information while dynamically ignoring distractor signals.

Importantly, the NoiseOut mechanism can be easily integrated with existing deep learning architectures by adding the gating mechanism and noise injection to the hidden layer before the classification layer while incurring only a small computational cost. This additional information processing step is inspired by the thalamus which modulates input stimuli before passing on the gated information to the prefrontal cortex for task completion.

Our experiments on the MNIST-C and ImageNet-C benchmarks demonstrated the effectiveness of NoiseOut in increasing the robustness of neural networks. Our proposed method outperformed the baseline CNN and other state-of-the-art methods, while also harmonizing well with existing techniques. Furthermore, we demonstrated that augmenting the loss function with a binarized version of the Full Integrated Gradients can further improve the performance of NoiseOut on MNIST-C.

However, there are several limitations in NoiseOut. Firstly, we noticed that the performance improvement afforded by NoiseOut depends on the size of the hidden layer to which NoiseOut is added to (See Figure 8 in Appendix B.3). Secondly, even though adding the NoiseOut mechanism only increased computational cost marginally (by about 3%; see Appendix B.4), this is not the case for the NoiseOut FIG variant, because of the repeated forward and backward passes for empirically estimating the IG. Lastly, besides filtering out distractor signal, the thalamus has been shown to amplify task relevant signals to improve task completion Schmitt et al. (2017), though this is not included in our current proposal.

Nevertheless, NoiseOut is a cheap and easy to integrate technique for improving the robustness of deep neural networks to novel perturbations in input data, and its combination with other state-of-the-art methods can lead to even more robust models. Future work can explore the use of secondary neural networks to predict the target Full Integrated Gradients to reduce the computational cost of that variant.

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

# A  DATASETS

## A.1  MNIST-C

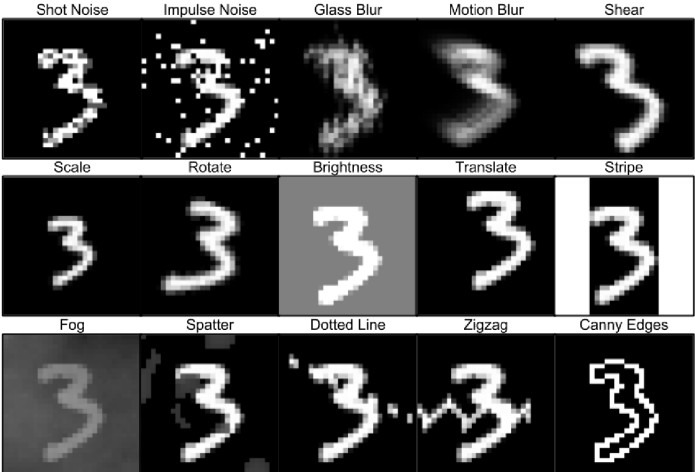

Figure 6: This Figure is taken from Mu & Gilmer (2019) and shows an example image for each of the perturbations in the test-set.

Figure 6 (taken from Mu & Gilmer (2019)) shows examples for each of the perturbations found in the MNIST-C test set.

## A.2  IMAGENET-C

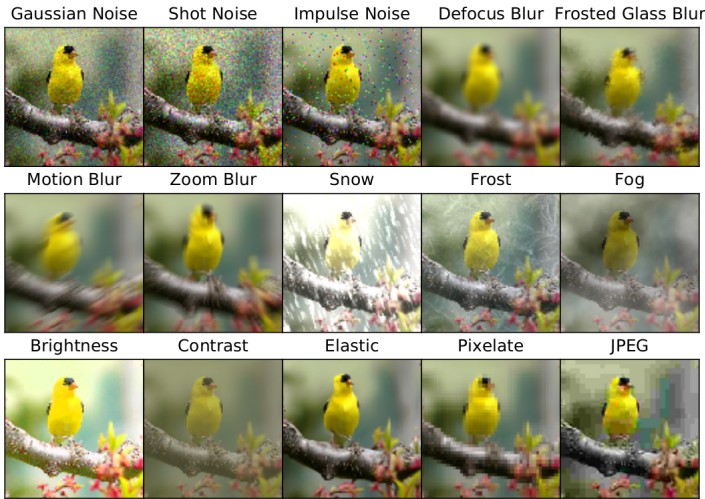

Figure 7: This Figure is taken from Hendrycks & Dietterich (2018) and shows an example image for each of the perturbations in the test-set.

Figure 7 (taken from Hendrycks & Dietterich (2018)) shows examples for each of the perturbations found in the ImageNet-C test set.

| NoiseInjection | 0% | 10% | 20% | 30% |
|---|---|---|---|---|
| NoiseOut FIG ($\lambda = 0.0$) | 91.09%$\pm$0.55 | 91.71%$\pm$0.35 | 92.03%$\pm$0.37 | 89.89%$\pm$0.46 |
| NoiseOut FIG ($\lambda = 0.5$) | **91.51%**$\pm$0.47 | **92.05%**$\pm$0.39 | **92.30%**$\pm$0.40 | 92.19%$\pm$0.43 |
| NoiseOut FIG ($\lambda = 1.0$) | 90.93%$\pm$0.47 | 91.90%$\pm$0.47 | 92.25%$\pm$0.31 | 92.42%$\pm$0.30 |
| NoiseOut FIG ($\lambda = 1.5$) | 90.53%$\pm$0.46 | 91.51%$\pm$0.47 | 92.22%$\pm$0.36 | 92.44%$\pm$0.26 |
| NoiseOut FIG ($\lambda = 2.0$) | 89.63%$\pm$1.26 | 91.39%$\pm$0.53 | 91.82%$\pm$0.46 | **92.46%**$\pm$0.24 |

Table 3: The table holds the performances achieved by NoiseOut FIG with different Noise Injection probabilities and different values for $\lambda$ (see Equation 9). The best performance for each Noise Injection percentage (column) is higlighted in bold. Each method was run 10 times and the results in the table correspond to the average and standard deviation of those runs

## B ADDITIONAL ANALYSIS

### B.1 THE $\lambda$ HYPERPARAMETER

As can be seen in Table 3, using $\lambda = 0.5$ produces the best result most often, whilst $\lambda = 2.0$ with a Noise Injection probability of 30% produces the best results overall. We hypothesize that this because it is hard for models to learn the correct patter (of what to gate) when too much Noise and too little signal is available. The additional FIG loss provides a secondary target to focus on in those cases. Since the $\lambda$ hyperparamter defines how much attention is paid to this additional loss term, it makes sense that for a high Noise Injection probability, a high $\lambda$ works best. It is unclear, however, why works best overall.

Since $\lambda = 0.5$ performs well more consistently, and because we don't want to present results with cherry-picked hyperparameters, we use $\lambda = 0.5$ for all experiments involving the FIG loss.
It is worth pointing out that NoiseOut FIG ($\lambda = 0.0$) is equivalent to the standard NoiseOut.

### B.2 NOISE INJECTION ALTERNATIVES

| Model (Noise pct.) | Additive | Uniform | Median Blur | Layer Injection | Noise Injection |
|---|---|---|---|---|---|
| CNN (0%) | 87.43%$\pm$1.29 | 87.43%$\pm$1.29 | 87.43%$\pm$1.29 | 87.43%$\pm$1.29 | 87.43%$\pm$1.29 |
| CNN (10%) | 88.44%$\pm$0.83 | 19.03%$\pm$12.06 | 87.42%$\pm$0.20 | **89.51%**$\pm$0.21 | 87.39%$\pm$0.85 |
| CNN (20%) | 87.70%$\pm$0.51 | 10.87%$\pm$0.41 | 88.99%$\pm$0.10 | **89.35%**$\pm$0.15 | 87.51%$\pm$1.45 |
| CNN (30%) | 87.08%$\pm$0.06 | 19.86%$\pm$6.02 | **88.31%**$\pm$0.57 | 71.31%$\pm$3.03 | 86.07%$\pm$1.53 |
| NoiseOut (0%) | 91.09%$\pm$0.55 | 91.09%$\pm$0.55 | 91.09%$\pm$0.55 | 91.09%$\pm$0.55 | 91.09%$\pm$0.55 |
| NoiseOut (10%) | 91.25%$\pm$0.13 | 87.50%$\pm$0.34 | 91.00%$\pm$0.08 | **91.84%**$\pm$0.14 | 91.71%$\pm$0.35 |
| NoiseOut (20%) | 90.69%$\pm$0.15 | 74.47%$\pm$0.55 | 91.07%$\pm$0.23 | **92.06%**$\pm$0.09 | 92.03%$\pm$0.37 |
| NoiseOut (30%) | **91.15%**$\pm$0.06 | 65.19%$\pm$5.52 | 90.78%$\pm$0.31 | 89.60%$\pm$0.44 | 89.89%$\pm$0.46 |
| NoiseOut FIG (0%) | 91.51%$\pm$0.47 | 91.51%$\pm$0.47 | 91.51%$\pm$0.47 | 91.51%$\pm$0.47 | 91.51%$\pm$0.47 |
| NoiseOut FIG (10%) | 91.24%$\pm$0.02 | 91.78%$\pm$0.02 | 91.54%$\pm$0.18 | **92.14%**$\pm$0.15 | 92.05%$\pm$0.39 |
| NoiseOut FIG (20%) | 91.58%$\pm$0.08 | 88.46%$\pm$0.85 | 91.41%$\pm$0.10 | 91.88%$\pm$0.19 | **92.30%**$\pm$0.40 |
| NoiseOut FIG (30%) | 91.42%$\pm$0.09 | 75.96%$\pm$2.55 | 91.39%$\pm$0.25 | 91.08%$\pm$0.14 | **92.19%**$\pm$0.43 |

Table 4: Testing all 3 models with different Noise strategies. Each method was run multiple times and the results in the table correspond to the average and standard deviation of those runs.

As can be seen in Table 4, the Additive, Median Blur, Layer Injection and Noise Injection Noise strategies all work similarly well, whilst Uniform Noise performs substantially worse, especially for a standard CNN classifier. Overall, Noise Injection (the method used for all experiments in Section 4, works best for NoiseOut FIG, and similarly well as Layer Injection for the standard NoiseOut.

For all of the above Noise strategies, we first randomly select a number of nodes in the specific layer (the number of nodes selected depends on the Noise percentage, denoted as the number in the parenthesis), and then edit these according to the strategy:

- **Uniform Noise**: Exchange the sampled Nodes with uniformly randomly generated numbers within the range of the smallest node and largest node in the current hidden states.

- **Additive Noise**: Add random numbers, generated from a normal distribution centered at 0 with standard deviation matching the empirical standard deviation in the hidden states, to the selected subset.

- **Median Blur**: For each selected node, take the Median of itself and the 10 neighboring nodes, and replace it with that value.

- **Layer Injection**: This is almost the same as Noise Injection, with the only difference that the Mean and Standard Deviation are determined in the current layer, rather than along the batch dimension.

### B.3 PERFORMANCE IMPACT OF VARIOUS HIDDEN STATE SIZES

To test what impact the number of hidden nodes that NoiseOut is applied upon has on the test performance, we randomly project the flattened final Conv layer to the required target size. As can be seen in Figure 8, the test performance correlates strongly with the number of nodes in the hidden layer that NoiseOut is applied to, especially when the number of nodes is smaller than 500. Afterwards, the relationship flattens out.

From these experiments it seems that the optimal number of hidden nodes NoiseOut is applied to is $50 * \#$ Classes. Thus, the theoritcal optimum for the ImageNet experiments would be $50,000$ (which is substantially more than the $2048$ the ResNet-50 architecture we are required to use for the ImageNet-C benchmark has).

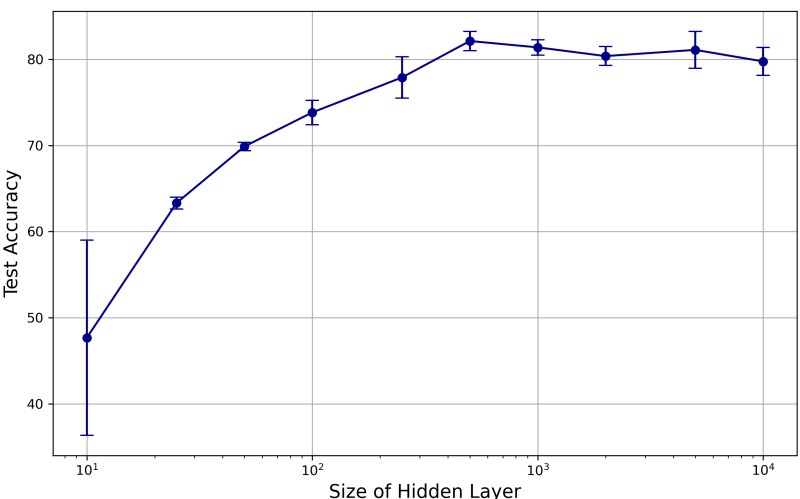

Figure 8: Test Accuracy vs Hidden Layer Size

### B.4 COMPUTATIONAL COST OF NOISEOUT

We measure the computational cost of adding NoiseOut by comparing the training time of the CNN without NoiseOut and with NoiseOut. Though this is not a perfect estimation, it has recently become a popular metric Geiping & Goldstein (2022). Table 5 shows the average time taken to train both models for 14 epochs on the clean MNIST train set. The experiments were run on a Nvidia RTX 3090.

As can be seen in Table 5, the training time increase when adding NoiseOut to an architecture is marginal (just over 3%).

| Model (Noise pct.) | Mean | Std. |
|---|---|---|
| CNN | 230.76 | 0.77 |
| NoiseOut (20%) | 237.91 | 1.96 |

Table 5: The table shows the time taken (in seconds) to train the models for 14 epochs on the MNIST train set.

## C ARCHITECTURE

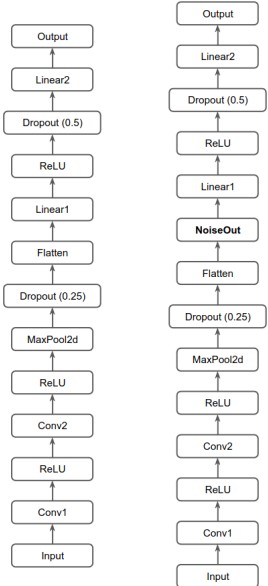

Figure 9: A diagram of the CNN (left) and NoiseOut augmented CNN (right) used for the MNIST-C experiments.

Figure 9 shows the diagrams for the CNN model and the NoiseOut augmented CNN models used in the MNIST-C experiments. It highlights well how the NoiseOut block seen in Figure 3 fits into an architecture. For the ImageNet experiments are ResNet-50 backbone is used, and the NoiseOut block is inserted directly after the flattening (same as in Figure 9).

## D COMPUTE USED

The MNIST-C experiments were run on smaller consumer GPUs (mainly Nvidia RTX 3080), whilst the ImageNet-C experiments required more vRam and were thus run on a Nvidia RTX 4090).

