# OpenReview forum: "NoiseOut: Learning to Gate Improves Robustness in Deep Neural Networks"
_ICLR.cc/2024/Conference — ICLR 2024 Conference Withdrawn Submission_

### Official Review · Reviewer_ZBE7 · 2023-10-31

**Soundness:** 2 fair
**Presentation:** 3 good
**Contribution:** 1 poor
**Rating:** 3
**Confidence:** 3

**Summary:**

This paper presents a NoiseOut gating unit for CNNs, trained to filter out distracting information that may lower the classifier's performance. The gating module is a one layer mlp which receives the hidden layer representations and applies the sigmoid function to estimate how much a signal is relevant or noisy. During training, random noise is injected by replacing a subset of the hidden states. The gating unit is supervised by an additional objective function based on the integrated Gradients approach to facilitate learning of the gates for suppressing the injected noise. The authors present results on MNIST-C and ImageNet-C datasets.

**Strengths:**

The paper is well-written and easy to follow. The idea of using a gating module to detect and suppress noisy signals is sound. The analysis that shows the gating activity is clustered for different perturbation types is interesting.

**Weaknesses:**

I have several concerns about the proposed method, mainly concerning evaluations and fairness of comparisons.

- Empirical Evaluation: The section detailing empirical evaluations appears notably insufficient. The paper lacks a comparative analysis with any SOTA robustness methods on the MNIST-C dataset. The authors mention the failure of a “plethora of state-of-the-art robustness methods” without providing detailed comparative data or analysis.

- Results on ImageNet-C: The paper does not present standalone results of the NoiseOut method on the ImageNet-C dataset. Results are only shown when the method is combined with others, yet claims of outperforming competing methods are made. This absence of direct comparison raises questions about the independent effectiveness of NoiseOut.

- Method Generalization: The evaluation of the method uses only two types of CNN architectures: a simple CNN (LeNet) for MNIST and ResNet50 for ImageNet. It remains unclear how well the proposed method generalizes across different or more complex architectures, and whether it can be scaled effectively.

- Gating Mechanism Analysis: There is a lack of comprehensive ablation studies on the gating mechanism’s behavior in response to various noise types. Insight into the failure cases, and an evaluation of how the noise sampling approach used during training addresses different noise categories (such as blurring, fog, frost, etc.) are critical. A detailed analysis, e.g. a comparison with median blur corruption strategies, would significantly strengthen the understanding and applicability of the proposed method.

- Compute and Parameter Cost Analysis: The paper does not discuss the computational and parameter costs associated with integrating the gating units. Appendix Figure 8 suggests a high optimal number of hidden nodes for NoiseOut application, potentially leading to significant computational overhead. Clarification is needed on whether this increase in parameters contributes to the notably higher performance at a 0% noise level on the MNIST dataset, compared to the baseline.

**Questions:**

See the weakness section.

---

### Official Review · Reviewer_zW13 · 2023-11-01

**Soundness:** 2 fair
**Presentation:** 3 good
**Contribution:** 2 fair
**Rating:** 3
**Confidence:** 5

**Summary:**

This paper introduces NoiseOut, a novel modular gating mechanism that aims at enhancing the robustness of DNNs to image perturbations. The authors use the Integrated Gradients method to identify distractor features causing classification errors and then propose NoiseOut to filter out these distractors. During training on clean datasets, they randomly replace hidden states with normal random values and integrate this into an additional objective function, allowing NoiseOut to learn dynamic gating policies. When tested on perturbed datasets, NoiseOut effectively filters out problematic features, thus improving DNN robustness, as demonstrated by strong results on the MNIST-C and ImageNet-C benchmarks.

**Strengths:**

The idea of NoiseOut is interesting and seems novel at least from my knowledge. It adds an attention-like module to the hidden states. The parameters are trained by an auxiliary loss basically saying that if the integrated gradient is negative, the weight should be zero, thus creating this denoising effect. The authors also intentionally add noise to the training by resetting some features to random noise, which could contribute to training. Overall, I think the idea is well-inspired and makes sense.

The experiments are also well-designed. The authors evaluate the method on two benchmark datasets focusing on different types of corruption. The authors also try to put their method on the top of some SOTA to demonstrate its effectiveness.

Overall, the presentation of the method and results are good and comprehensive.

**Weaknesses:**

The main weakness I can see in the methodology is that this method may be over-engineering the representation noise. Deep networks has been shown to be robust to a lot of challenging cases due to their over-parameterization nature. It is likely that a well-trained model is actually robust to the noise such that those noises won't really affect the deep features if there is enough augmentation, etc. in the training process. The proposed module also requires an auxiliary loss, which may add more effort to hyperparameter tuning and thus more difficult to get actual better results. The results on ImageNet-C somehow prove this, as if there exists a strong augmentation pipeline (DeepAugment + AugMix), the proposed NoiseOut can hardly improve the results further.

More specific weaknesses and questions beyond the general idea:
1. I wonder how NoiseOut works with batch norm, does it only work with layers without batch norm?
2. Does adding manual noise in training actually improve the performance of NoiseOut? How does it compare to general data augmentation methods?
3. A justification of the effect of different lambda values is needed.
4. The architectures used in the paper are outdated. To better demonstrate the effectiveness, the authors should use some more recent architectures such as EfficientNet (or other lightweight models) and ViT (or other large models).
5. The ImageNet results show only marginal improvement. Statistical tests with more trials are needed to justify the improvement.
6. I'm not sure what the LDA analysis is trying to highlight. Is there a connection between dynamic gating policy and the model's robustness?

**Questions:**

See weakness.

---

### Official Review · Reviewer_G3rD · 2023-11-02

**Soundness:** 2 fair
**Presentation:** 2 fair
**Contribution:** 2 fair
**Rating:** 3
**Confidence:** 4

**Summary:**

The paper proposes a method called NoiseOut, which is a lightweight modular gating mechanism that can be integrated with existing DNNs to enhance their robustness. It demonstrates that the proposed method achieves strong performance on two robustness benchmarks, MNIST-C and ImageNet-C, and outperforms or complements existing methods.

**Strengths:**

1.	The proposed lightweight modular gating mechanism can be easily integrated with existing deep learning models.
2.	The visualizations in this paper help us gain some understanding of the neural networks and the proposed gating mechanism.

**Weaknesses:**

1.	This paper needs a more extensive comparison to SOTA or existing methods for improving corruption robustness.
2.	The discussion of the limitations and potential applications of the proposed method is ignored. For example, what is the performance when the proposed method is used in the ViT architecture?
3.	The improvement of the proposed additional FIG loss term appears to be trivial, as shown in Table 1.

**Questions:**

1. What is the performance when the proposed method is used in the ViT architecture?
2. What is the performance of NoiseOut alone on ImageNet-C?
3. Has the resource consumption and time cost of the proposed method been compared?

---

### Official Review · Reviewer_oCeg · 2023-11-03

**Soundness:** 2 fair
**Presentation:** 3 good
**Contribution:** 2 fair
**Rating:** 3
**Confidence:** 3

**Summary:**

This paper propose a method named NoiseOut to improve corruption robustness of CNNs. The two key points of the method are: during training on clean images, randomly replace a subset of hidden states with normally sampled values, and use an additional objective function to minimize the L2 distance between the activated neurons and the binarized full integrated gradients. Experiments on MNIST-C and ImageNet-C on CNN backbones show improvements upon previous methods.

**Strengths:**

Improving corruption robustness is an important problem. The authors seek inspiration from biological systems and propose to learn a gating mechanism to filter out irrelevant information for classification. The proposed method is orthogonal to previous methods such as data augmentation and thus can be combined with existing methods.

**Weaknesses:**

Despite the seemingly promising example from biological systems, I am not fully convinced by the need of gate information presented in section 3.1.
- First of all, I do not necessarily understand why we should expect the features corresponding to the corrupted images to have more negative FIG values than those of the clean images. In the integrated gradient paper, their sensitivity axiom is: If the function implemented by the deep network does not depend (mathematically) on some variable, then the attribution to that variable is always zero. So if the corrupted image contains more irrelevant features, shouldn't it have more zero IG values rather than negative IG value?
- Second, I do not think the brightness example in Figure 1 is a good demo for corruption robustness. To me, the bright seven is also a seven, and contains no less information about seven than the "clean" image. An Imagenet image may be a better motivating example.
- Third, a histogram on the FIG of a single image does not really make thee point that corrupted images really have more negative FIG values. More quantitative results are needed to support that statement.

I am not very convinced by the scalability of this method.
- The experiments on MNIST-C is mostly an ablation study and compare with several variants based on the proposed method.
- The experiments on ImageNet-C show slight improvement upon previous method (0.2%). Without an error bar, it is hard to see that this is a salient improvement.

**Questions:**

My main confusion is why we should inhibit negative FIG values since the features of negative FIG values should still be relevant for the classification.

Since ViTs have been shown to have better robustness than CNNs. Does the proposed method work on transformer architectures?